# Transforming recursive programs for parallel execution

**Alexey Radul**    **Brian Patton**    **Dougal Maclaurin**    **Matthew D. Hoffman**

**Rif A. Saurous**
Google AI, { axch, bjp, dougalm, mhoffman, rif } @google.com

## Abstract

We present a general approach to batching arbitrary computations for accelerators such as GPUs. We show orders-of-magnitude speedups using our method on the No U-Turn Sampler (NUTS), a workhorse algorithm in Bayesian statistics. The central challenge of batching NUTS and other Markov chain Monte Carlo algorithms is data-dependent control flow and recursion. We overcome this by mechanically transforming a single-example implementation into a form that explicitly tracks the current program point for each batch member, and only steps forward those in the same place. We present two different batching algorithms: a simpler, previously published one that inherits recursion from the host Python, and a more complex, novel one that implements recursion directly and can batch across it. We implement these batching methods as a general program transformation on Python source. Both the batching system and the NUTS implementation presented here are available as part of the popular TensorFlow Probability software package.

## 1 Introduction

Modern machine learning accelerators such as GPUs are oriented around Single Instruction Multiple Data (SIMD) parallelism—doing the same thing to each item of a big array of data at once. Machine learning programs optimized for such accelerators generally consist of invoking *kernels*, where each kernel is a separately hand-tuned accelerator program for a specific function. Good utilization of the accelerator comes of making relatively few kernel calls, with each kernel processing a relatively large amount of data. In the case of a typical neural network workload, the kernels would be "matrix multiplication" or "convolution", and the call sequence would encode the architecture of the neural network.

Let's briefly look at the resulting programming model. This review is worded in the TensorFlow [1] ecosystem, since that's the setting for our work, but other machine learning frameworks are broadly similar. The top-level program is generally written in Python, calling TensorFlow API functions that correspond to kernels such as matrix multiplication. These functions can be executed immediately, in the so-called TensorFlow *Eager mode*. In this case they can be arbitrarily interleaved with the host Python, including control flow; but suffer corresponding dispatch and communication overhead. Alternately, the same API functions can be used to construct an operation graph to be executed all at once. This is the so-called TensorFlow *graph mode*. The advantage is that graphs can be saved, loaded, and optimized before being run, and suffer less dispatch overhead. The disadvantage is that graph computations cannot be interleaved with the host Python, and in particular graph mode cannot represent recursive computations. A third option is to further compile the graph with XLA [20]. XLA imposes even more restrictions, such as statically resolving the shapes of all intermediate arrays, but offers the additional benefit of fusing kernels together, which reduces dispatch overhead even more.

Good performance in this programming style depends heavily on vectorization, both within the kernels and at the level of kernel inputs. One very common strategy for vectorizing machine learning programs is so-called *batching*: processing a batch of independent inputs in lock-step in order to get more play for vectorization. Batching can also reduce per-input memory pressure: in the case of a neural network with $N$ features, each input has size $O(N)$, whereas the weight matrices can easily have size $O(N^2)$. Running multiple inputs through the layers of the network in lock-step can re-use each weight matrix for many examples before having to evict it from memory caches in order to load the next one.

It is standard practice in machine learning frameworks such as TensorFlow or PyTorch [15] to code the kernels to accept extra input dimensions and operate elementwise across them. Consequently, coding a batched version of a straightline program is relatively straightforward, if somewhat tedious and error-prone. Simple neural networks being straightline, batch training is the norm. Obstacles arise, however, when one wishes to batch a program with control flow, such as conditionals or variable-length loops. Then it becomes necessary to keep track of which batch member takes which branch of each conditional, and avoid or ignore computations on batch members at the wrong program point. The difficulty of doing this by hand impedes using sophisticated classical algorithms in machine learning. Despite the impedance, people have used tree searches [18], optimization routines [4] and ordinary differential equations solvers [7] in machine learning work; what else could we accomplish if it were easier?

Additional obstacles arise when trying to run a recursive program on a modern machine learning framework, in batch or otherwise, because the dataflow graph representation cannot execute recursion natively. This is as true in XLA or TensorFlow graph mode as it is in other graph-oriented machine learning frameworks like Caffe [9]. The user is therefore forced to fall back to eager-style execution, paying more communication overhead. If machine learning is to benefit fully from the last 60 years of computer algorithm development, we must be able to run recursive algorithms reasonably efficiently.

Our goal in this paper is to push the boundary of what classical algorithms can efficiently execute on accelerators, in the context of modern machine learning frameworks. In particular, we

- Introduce *program-counter autobatching* (Section 3), a global, static program transformation for batching programs with arbitrary control flow, and materializing recursion into an underlying dataflow system.
- Demonstrate that program-counter autobatching can successfully accelerate the No U-Turn Sampler, a classic algorithm from Bayesian statistics, by compiling its recursion into explicit stack management, and by statically constructing a schedule for running it on batches of inputs.
- Describe, using the same vocabulary, *local static autobatching* (Section 2). This is a simpler and lower-overhead batching transformation with less batching power in the recursive case.
- Directly compare these two autobatching strategies on a test problem from Bayesian statistics (Section 4).

Program-counter autobatching is available as a module in the popular TensorFlow Probability [19, 8] software package. That module also implements a local static autobatching variant for comparison.

## 2 Local Static Autobatching

The simplest batching strategy (whether automated or hand-coded) is to retain the graph of the computation as-is and just transform every operation into a batched equivalent. We call this *local static autobatching*. Intuitively, it's "local" because the pattern of composition of operations doesn't change, and every operation can be transformed on its own; and it's "static" because the batching schedule doesn't depend on the input data, and can thus be computed before starting execution.

When extending this idea to programs with control flow, it is necessary to at least introduce a mask of which batch members are "currently active". One then arranges to execute every control path that at least one batch member follows, and avoid or ignore each computation for each batch member that did not take that path. If the program being batched is recursive, the recursion still has to be carried out by the control language, i.e., Python. This transformation can be implemented in many styles [5, 2, 6]. We choose to implement local static autobatching here as a static program transformation

targeting a simple control flow language. In this language, all the user's actual computations become Primitive operations, and the control and recursion constructs are encoded in a standard way in Jump, Branch, Call, and Return instructions. The actual batching is accomplished by a nonstandard interpretation of that language.

In addition to storage for all the batch member inputs, the runtime maintains an *active set* (initially the whole batch) and a *program counter* (initially the start of the entry point). The active set is a mask—all inactive batch members are ignored and never modified until they become active. The program counter gives the program point (as a basic block index) each active batch member is waiting to execute. The execution model is simple: at each step, we select some basic block that has at least one active batch member and execute it in batch. We then update the data storage and program counters of just those *locally active* batch members. Repeat until all active batch members have exited the function, then return. If the block we are executing ends in a branch (i.e., the prelude of a source language `if` statement), the locally active batch members may *diverge*, in that some may move to the true branch and some to the false. They will *converge* again when both of those branches complete, and we continue after the end of the `if`. If the block we are executing contains a (potentially recursive) call to a function the user asked us to auto-batch, we appeal to the host language's function call facility. The only trick is to update the active set in the recursive autobatching invocation to include only the locally active batch members (i.e., those whose program counter was at that call).

## 3 Program Counter Autobatching

The local static autobatching discussed in Section 2 has an interesting limitation. Because it relies on the Python stack to implement recursion, it cannot batch operations across different (recursive) calls to the same user function. So two batch members could be trying to execute the same code and not be batchable because the system doesn't have a long-enough sight line to connect them. And, of course, relying on Python to manage the recursion imposes communication costs and limits the optimizations the underlying machine learning framework can do.

We can serve two purposes with one intervention by implementing the stack within the autobatching system. We choose to give each program variable its own stack (by extending the relevant array with another dimension). To implement this, we want a slightly different control flow graph language. Since the runtime is now managing the stacks itself, we replace the Call instruction with explicit (per-variable) Push and Pop instructions, as well as PushJump for entering function bodies. The Push also computes the value to write to the top of the variable's stack. As compared with local static autobatching, the active set no longer persists across steps, while the explicit program counter takes on a more central role (hence the name). The program counter now has a stack dimension of its own. The locally active set can now include batch members at different stack depths, giving the runtime a chance to batch their computations together. Consequently, computations can converge by calling into the same subroutine from different code locations, and conversely diverge by returning thereto. The major advantage is that the runtime is no longer itself recursive, so can be coded completely in a system like (graph-mode) TensorFlow or XLA that does not support recursion natively.

## 4 Experiments

We evaluate autobatching on the No U-Turn Sampler (NUTS), a workhorse gradient-based Markov-chain Monte Carlo method widely used in Bayesian statistics. We chose NUTS as a test target because (i) the standard presentation is a complex recursive function, prohibitively difficult to batch by hand, and (ii) batching across multiple independent chains can give substantial speedups. Our results show that batching on GPUs enables NUTS to efficiently scale to thousands of parallel chains, and get inferential throughput well beyond existing CPU-based systems such as Stan. By demonstrating performance gains from batching NUTS, we hope to contribute to a broader practice of running large numbers of independent Markov chains, for more precise convergence diagnostics and uncertainty estimates.

We test autobatched NUTS on inference in a Bayesian logistic regression problem with 10,000 synthetic data points and 100 regressors. We test three forms of autobatching NUTS:

- Program counter autobatching, compiled entirely with XLA;

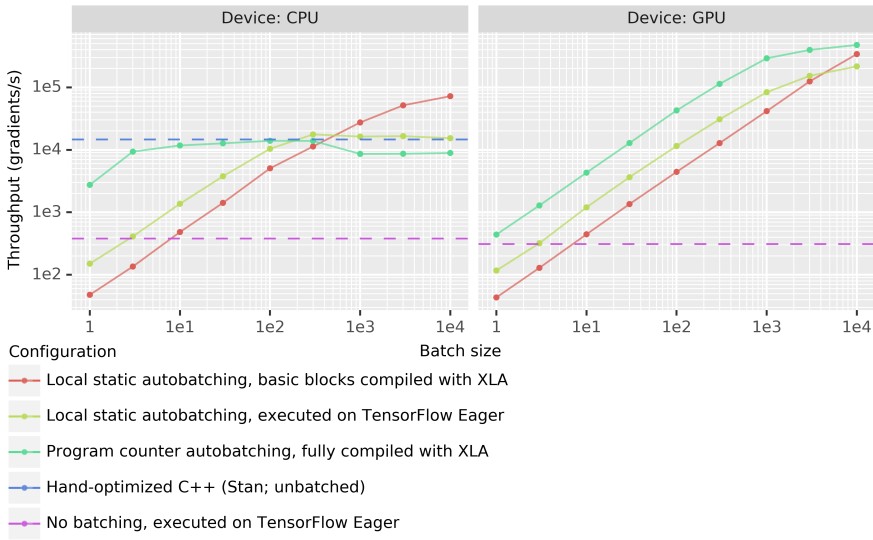

Figure 1: Performance of auto-batched No U-Turn Sampler on the Bayesian logistic regression problem (100 latent dimensions, 10,000 data points). The batch size refers to the number of chains running in tandem. The reported gradients are the total across all chains, excluding waste due to synchronization. We compare the performance of program counter autobatching compiled with XLA to our local static autobatching executed in TensorFlow's Eager mode. We also include two baselines. One is the same program executed directly in Eager mode without autobatching (perforce running one batch member at a time). The other is the widely used and well-optimized Stan implementation of (a variant of) the same NUTS algorithm. Batching provides linear scaling on all tested platforms, until the underlying hardware saturates. See text for details of the experimental setup.

- Local static autobatching, executed entirely with TensorFlow Eager; and
- A hybrid: Running the control operations of local static autobatching in TensorFlow Eager, but compiling the straight-line components (basic blocks) with XLA.

The purpose of the latter is to try and tease apart the benefits of compilation specifically for control flow versus compiling (and fusing together) straightline sequences of kernel invocations. Note that identifying the basic blocks to compile separately is a nontrivial program transformation in its own right, which our software framework handles naturally.

In Figure 1, we measure the number of model gradient evaluations per second we can get out of batching NUTS in different ways. The main effect we see is GPU throughput scaling linearly with batch size. We also see the speedup from avoiding cycling to Python (on the host CPU!) to implement the recursion. The behavior when running entirely on CPU is more nuanced. CPUs are superb at control flow and recursion as it is, so the main effect of batching seems to be to amortize away platform overhead, until we match the performance of the Stan system's long-optimized custom C++ at a batch size of a few hundred—or just ten for compiling fully with XLA, whose per-leaf-kernel overhead is much smaller.

A few details of the experimental setup. These measurements are for the synthetic Bayesian logistic regression problem. The measured time counts only a warm run, excluding compilation, the one-time TensorFlow graph construction, etc. This allows measurements for small batch sizes not to be swamped by $O(1)$ overhead. The CPU computations were run on a shared 88-core machine with 100 GiB of RAM allocated to the benchmark, in 32-bit floating-point precision. The GPU computations were run on a dedicated Tesla P100 GPU, also in 32-bit precision. In all cases, we slightly modified the published NUTS algorithm to take 4 steps of the leapfrog integrator at each leaf of the NUTS tree, to better amortize the control overhead. This has no effect on the soundness of the algorithm. The timings are best of five independent runs. Due to technical limitations, the Stan baseline was run on different hardware. We scaled its throughput against a calibration run of program counter autobatching on the same machine and precision.

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
