# OpenReview forum: "Transforming recursive programs for parallel execution"
_NeurIPS.cc/2019/Workshop/Program_Transformations — Program Transformations @NeurIPS2019 Poster_

### Official Review · AnonReviewer1 · 2019-09-26
**A source transformation plus an execution system for increased efficiency of "batch computation" on GPUs. Good**

**Confidence:** 4
**Rating:** 6

**Review:**

Paper describes a framework to increase performance of batches of computations when executed on e.g. GPU's. The idea is to reduce the overhead by exchanging the parallel level (the iteration over batch elements) and the control-flow level (the program call graph and flow-graph), by pushing the parallel level down under the flow-graph level and even the call-graph level (even when recursive). Requires several technical manipulations including a program transformation. Appears also to require a complex support system with an emulated execution stack. Obtained speedup appears quite good on some cases. More testing is probably on the way.
Serious and correct paper.
A part of the paper is really about program transformation, although the target form is probably complex.
There doesn't seem to be a link to the particular question of gradients.
Agreeing with the author's reply: this abstract is nevertheless in the scope of the workshop, so rating was modified accordingly. However the idea presented appears to require more testing and validation so far.

---

### Official Review · AnonReviewer2 · 2019-09-30
**Static forms of autobatching**

**Confidence:** 4
**Rating:** 7

**Review:**

Targeting accelerators such as GPUs crucially depends on vectorization for performance, often achieved by batching. However, batching is notoriously difficult in the presence of (divergent) control flow. This paper presents a new approach called program-counter autobatching which materializes recursion schemes  into an underlying data-flow system. The authors claim that the approach can batch programs with arbitrary control flow, and they demonstrate some pretty impressive speedups on a classic sampling algorithm from Bayesian statistics (no-U-turn sampler).

While the work seems solid, what prevents me from being even more enthusiastic is that there is only a single experiment, and that the paper doesn’t even try to frame the contribution within the appropriate context of several decades of compiler research on automatic  parallelization of irregular workloads.

---

### Decision · Program_Chairs · 2019-10-01

**Decision:**

Accept (Poster)

**Comment:**

The reviewers argued that this is a strong contribution, but that it lacks experimental validation and should be related more strongly to existing literature on compiler optimizations.